# SOCS3 Ablation in Leptin Receptor-Expressing Cells Causes Autonomic and Cardiac Dysfunctions in Middle-Aged Mice despite Improving Energy and Glucose Metabolism

**DOI:** 10.3390/ijms23126484

**Published:** 2022-06-10

**Authors:** João A. B. Pedroso, Ivson B. da Silva, Thais T. Zampieri, Leonardo T. Totola, Thiago S. Moreira, Ana P. T. Taniguti, Gabriela P. Diniz, Maria Luiza M. Barreto-Chaves, Jose Donato

**Affiliations:** 1Departamento de Fisiologia e Biofisica, Instituto de Ciencias Biomedicas, Universidade de Sao Paulo, Sao Paulo 05508-000, Brazil; nutri.pedroso@gmail.com (J.A.B.P.); thaizampieri@gmail.com (T.T.Z.); ltotola@ig.com.br (L.T.T.); tmoreira@icb.usp.br (T.S.M.); 2Departamento de Anatomia, Instituto de Ciencias Biomedicas, Universidade de Sao Paulo, Sao Paulo 05508-900, Brazil; ivsonbs@hotmail.com (I.B.d.S.); aptiemi@yahoo.com.br (A.P.T.T.); gpdiniz@usp.br (G.P.D.); mchaves@usp.br (M.L.M.B.-C.); 3Departamento de Morfologia, Centro de Ciencias da Saude, Universidade Federal da Paraiba, Joao Pessoa 58059-900, Brazil

**Keywords:** aging, cardiovascular system, heart, hypothalamus, leptin, obesity

## Abstract

Leptin resistance is a hallmark of obesity. Treatments aiming to improve leptin sensitivity are considered a promising therapeutical approach against obesity. However, leptin receptor (LepR) signaling also modulates several neurovegetative aspects, such as the cardiovascular system and hepatic gluconeogenesis. Thus, we investigated the long-term consequences of increased leptin sensitivity, considering the potential beneficial and deleterious effects. To generate a mouse model with increased leptin sensitivity, the suppressor of cytokine signaling 3 (SOCS3) was ablated in LepR-expressing cells (LepR^∆SOCS3^ mice). LepR^∆SOCS3^ mice displayed reduced food intake, body adiposity and weight gain, as well as improved glucose tolerance and insulin sensitivity, and were protected against aging-induced leptin resistance. Surprisingly, a very high mortality rate was observed in aging LepR^∆SOCS3^ mice. LepR^∆SOCS3^ mice showed cardiomyocyte hypertrophy, increased myocardial fibrosis and reduced cardiovascular capacity. LepR^∆SOCS3^ mice exhibited impaired post-ischemic cardiac functional recovery and middle-aged LepR^∆SOCS3^ mice showed substantial arhythmic events during the post-ischemic reperfusion period. Finally, LepR^∆SOCS3^ mice exhibited fasting-induced hypoglycemia and impaired counterregulatory response to glucopenia associated with reduced gluconeogenesis. In conclusion, although increased sensitivity to leptin improved the energy and glucose homeostasis of aging LepR^∆SOCS3^ mice, major autonomic/neurovegetative dysfunctions compromised the health and longevity of these animals. Consequently, these potentially negative aspects need to be considered in the therapies that increase leptin sensitivity chronically.

## 1. Introduction

Obesity is a complex and multifactorial disease caused by a chronic imbalance between food intake and energy expenditure [1,2]. Obesity is considered a pandemic since a high percentage of people in numerous developed and developing countries present excessive accumulation of body fat [3,4]. Obesity increases the risk of multiple health problems, which ultimately lead to a significant decrease in quality of life, productivity, and life expectancy [1,2]. Thus, the development of efficient and safe approaches that can prevent and treat this metabolic disease are urgently needed.

Leptin resistance is a hallmark of obesity [5,6,7]. Obese individuals exhibit high leptin concentrations, which do not result in the expected suppression of food intake and increased energy expenditure [5,6,7]. Additionally, obese subjects in general do not respond to exogenous leptin treatment [8,9]. Consequently, leptin administration is inefficient for the treatment of obesity. In contrast, drugs aiming to increase the sensitivity to leptin are among the most promising pharmacological alternatives to treat obesity [6,7,10,11,12,13,14]. For example, previous studies have shown a synergistic interaction between amylin and leptin to reduce body weight [12,13]. Other studies have demonstrated that either celastrol or withaferin A are leptin sensitizers that are able to induce weight loss [10,11]. Furthermore, several plant-derived natural products may present leptin sensitizing properties by inhibiting molecules that induce leptin resistance, such as protein-tyrosine phosphatase 1B (PTP1B) [15]. Thus, drugs that increase leptin sensitivity in theory could take advantage of the anorexigenic action of the endogenously produced hormone to cause persistent weight loss.

Central leptin action not only regulates food intake and energy expenditure but also modulates several neurovegetative and autonomic functions, including the cardiovascular system and hepatic gluconeogenesis [7]. In this regard, central leptin administration or leptin action on pro-opiomelanocortin (POMC)-expressing neurons strongly modulates hepatic insulin sensitivity and suppresses glucose production [16,17,18]. Conversely, leptin action on neurons of the dorsomedial nucleus of the hypothalamus (DMH) regulates the sympathetic tone, heart rate (HR) and blood pressure [19,20]. Therefore, leptin receptor (LepR) signaling in DMH neurons is likely involved in the pathophysiology of hypertension that is associated with obesity.

Importantly, different neuronal populations exhibit a distinct predisposition to develop leptin resistance. While neurons of the arcuate nucleus of the hypothalamus frequently develop insensitivity to leptin in obese animals, other neuronal populations, including DMH neurons, remain responsive to leptin even in hyperleptinemic obese individuals [20,21]. The hyperactivity of leptin in the neurons that retain leptin sensitivity may produce long-term deleterious effects, such as favoring an increased sympathetic tone that leads to hypertension and other cardiovascular problems [19,20]. Thus, although therapies that increase leptin sensitivity may present clear beneficial effects in the regulation of energy balance, the potential negative aspects of such approaches have been poorly examined. In the present study, we investigated the long-term consequences of increased leptin sensitivity in mice. For this purpose, we used a mouse model carrying ablation of the gene encoding the suppressor of cytokine signaling 3 (SOCS3), which is a well-known protein that inhibits LepR signaling, its expression is increased in the hypothalamus of obese animals, and whose genetic deletion leads to increased leptin sensitivity and metabolic improvements [5,6,7,22,23,24,25,26,27,28,29,30,31,32]. The potential beneficial and deleterious consequences of SOCS3 ablation in LepR-expressing cells were investigated in young adult and middle-aged male mice.

## 2. Results

### 2.1. SOCS3 Ablation in LepR-Expressing Cells Improves Energy and Glucose Homeostasis in Aging Mice

Considering the well-established role of the SOCS3 protein in inhibiting LepR signaling [5,6,7,22,23,24,25,26], we produced mice carrying SOCS3 deletion in LepR-expressing cells to investigate the long-term consequences of increased leptin sensitivity. This mouse model was characterized and validated in several previous studies [27,28,29,30,31,32]. For example, former studies have shown that SOCS3 deletion in LepR-expressing cells increases leptin sensitivity in pubertal and pregnant female mice [28,29,31], as well as in male mice consuming either chow or a high-fat diet [27,30]. Additionally, the ability of leptin, a high-fat diet or pregnancy to upregulate *Socs3* mRNA levels in the hypothalamus is completely prevented in LepR^∆SOCS3^ mice [27,28]. Initially, the changes in body weight over time were determined. Although the body weight of LepR^∆SOCS3^ mice showed no changes in the first months of life, these conditional knockout mice exhibited a lower weight gain over time that led to a decreased body weight in comparison with control littermate mice (Figure 1A). LepR^∆SOCS3^ mice also showed reduced body adiposity, which was confirmed by a decrease in the perigonadal fat pad and in the sum of the fat deposits (Figure 1B). To determine the possible causes of these metabolic differences, food intake and energy expenditure (VO_2_) were determined in young adult (3-month-old) or middle-aged (approximately 15-month-old) mice. LepR^∆SOCS3^ mice exhibited reduced food intake at a young age, but not in middle-aged animals (Figure 1C). No significant changes in energy expenditure were observed in LepR^∆SOCS3^ mice, although middle-aged animals showed reduced energy expenditure, compared to young adult mice (Figure 1D). No significant changes between the groups were observed in the respiratory quotient (Figure 1E). The ambulatory activity of middle-aged mice was reduced when compared to young adult animals. However, LepR^∆SOCS3^ mice did not exhibit differences as compared to control animals (Figure 1F). Therefore, SOCS3 ablation in LepR-expressing cells prevents weight gain during aging, likely by reducing food intake.

Then, possible changes in glucose homeostasis were evaluated. Although young adult LepR^∆SOCS3^ mice consistently showed reduced basal blood glucose concentrations, their glucose tolerance and insulin sensitivity were relatively similar to the control mice (Figure 2A,B). In contrast, middle-aged LepR^∆SOCS3^ mice displayed improved glucose tolerance and insulin sensitivity (Figure 2C,D). Thus, the absence of SOCS3 in LepR-expressing cells improves glucose homeostasis in aging mice.

### 2.2. Increased Leptin Sensitivity in Middle-Aged LepR^∆SOCS3^ Mice

Aging induces leptin resistance and increases hypothalamic SOCS3 expression [33,34,35,36,37,38,39,40,41]. Thus, we determined whether SOCS3 ablation in LepR-expressing cells is sufficient to prevent aging-induced leptin resistance. As expected, an acute systemic leptin injection (2.5 µg/g body weight) reduced the food intake of young adult control mice (Figure 3). A similar reduction in food intake was observed in young adult LepR^∆SOCS3^ mice (Figure 3). In contrast, middle-aged control mice did not exhibit the anorexigenic response to leptin, suggesting leptin resistance. Importantly, middle-aged LepR^∆SOCS3^ mice remained responsive to leptin (Figure 3). Thus, SOCS3 ablation in LepR-expressing neurons is sufficient to prevent aging-induced leptin resistance.

### 2.3. Cardiac Abnormalities and Decreased Survival Rate in Aging LepR^∆SOCS3^ Mice

Although middle-aged LepR^∆SOCS3^ mice exhibited improvements in several metabolic parameters (e.g., reduced body weight and improved glucose homeostasis), we observed frequent deaths in LepR^∆SOCS3^ mice over time. Thus, a survival analysis was performed and, remarkably, a very high mortality rate was observed in aging LepR^∆SOCS3^ mice (*p* < 0.0001; Log-rank test). For example, only approximately 20% of the conditional knockout mice reached more than 16 months of life (Figure 4A). To identify the causes of this unexpected mortality, we investigated the presence of possible cardiac abnormalities in the mutant mice. The relative cardiomyocyte diameter was significantly increased in both young adult and middle-aged LepR^∆SOCS3^ mice (Figure 4B,C), indicating cardiomyocyte hypertrophy. Furthermore, SOCS3 ablation in LepR-expressing cells caused an increase in the relative collagen area of the heart in both young adult and middle-aged mice (Figure 4D), which indicates the development of myocardial fibrosis.

To determine whether cardiomyocytes express the long-isoform LepR (LepRb), we used a well-established LepRb-reporter mouse [42,43]. Although the tdTomato reporter protein was observed in the extracellular matrix, LepRb expression was practically absent in cardiomyocytes (Figure 5A,C).

Subsequently, mean arterial pressure (MAP) and HR were determined. Although LepR^ΔSOCS3^ mice exhibited similar MAP (Figure 6A) and HR (Figure 6D) as compared to the control mice, the systolic arterial pressure (SAP) and pulse interval (PI) variabilities were significantly increased in LepR^ΔSOCS3^ mice (Figure 6B,E). Spectral analyses indicated an increase in the low-frequency (LF) SAP and PI variabilities in LepR^ΔSOCS3^ mice (Figure 6C,F), whereas the high-frequency (HF) PI variability remained unchanged (Figure 6G). Consequently, the sympathovagal balance was significantly increased in LepR^ΔSOCS3^ mice (Figure 6H). Considering the role played by the sympathetic nervous system regulating the LF variability of SAP and PI, and the increased LF/HF ratio [44,45,46,47], our findings suggest an increased sympathetic activity to the vasculature and heart in LepR^ΔSOCS3^ mice.

To evaluate how SOCS3 ablation in LepR-expressing cells may affect functional aspects of the cardiovascular system, mice were subjected to an incremental treadmill maximal running test. Only young adult animals were used in this test since similar histological alterations were observed between young adult and middle-aged LepR^∆SOCS3^ mice. We observed that LepR^∆SOCS3^ mice fatigued faster (Figure 4E) and presented reduced mean velocity during the maximal test (Figure 4F) compared to control animals. Therefore, SOCS3 ablation in LepR-expressing cells leads to a lower survival rate, cardiac remodeling and reduced cardiovascular capacity that are apparent even in young adult animals.

To further evaluate the cardiac function of LepR^∆SOCS3^ mice, the Langendorff technique of isolated heart perfusion was used [48]. Baseline cardiac function was assessed by measuring four different parameters during the stabilization time: left ventricular developed pressure (LVDP; mmHg), HR (bpm), the first derivative of the positive (+dP/dt; mmHg/s) and negative (−dP/dt; mmHg/s). LepR^ΔSOCS3^ mice did not present alterations in LVDP, HR, +dP/dt, and −dP/dt relative to the respective control groups in both young adult (Figure 7A–D) and middle-aged animals (Figure 8A–D). After the ischemia period, a temporal analysis of cardiac parameters at 15, 30, 45 and 60 min of reperfusion was evaluated. Functional cardiac recovery values were expressed as a percentage of baseline values (during stabilization) in each group. In two different times of reperfusion period (30 and 45 min), the LVDP percentage of recovery significantly decreased in young adult LepR^ΔSOCS3^ mice (Figure 7E), suggesting impairment of post-ischemic cardiac functional recovery. The HR, +dP/dt and −dP/dt did not differ between groups of young adult mice (Figure 7F–H).

When middle-aged animals were evaluated, LepR^ΔSOCS3^ mice presented an increase in LVDP at 30, 45 and 60 min of the reperfusion, when compared to the control group (Figure 8E). HR was reduced in middle-aged LepR^ΔSOCS3^ mice at 30 and 60 min of the reperfusion (Figure 8F). No differences between groups were found in the cardiac inotropism (+dP/dt) and lusiotropism (−dP/dt) during the reperfusion period (Figure 8G–H). Thus, in contrast to young adult LepR^ΔSOCS3^ mice, middle-aged LepR^ΔSOCS3^ mice exhibited an apparent improvement in the post-ischemic cardiac functional recovery. However, substantial arhythmic events were consistently observed in middle-aged LepR^ΔSOCS3^ mice during the reperfusion period (Figure 8I). Therefore, SOCS3 ablation in LepR-expressing cells causes several cardiac abnormalities, which can compromise the cardiovascular capacity and possibly the survival of aging mice.

### 2.4. LepR^∆SOCS3^ Mice Exhibit Fasting-Induced Hypoglycemia That Is Associated with Reduced Gluconeogenesis

To investigate whether LepR^∆SOCS3^ mice may present defects in the control of other neurovegetative functions, mice were subjected to 24 h or 48 h fasting to evaluate their ability to maintain glycemia, which is highly dependent on brain centers influenced by leptin action [49,50,51,52,53,54,55]. LepR^∆SOCS3^ mice showed lower blood glucose concentrations at the fed state (Figure 9A). Although 24 h fasting reduced the glycemia similarly in the control and the LepR^∆SOCS3^ mice, the conditional knockout mice exhibited hypoglycemia when fasting was maintained for 48 h (Figure 9A). After providing food to the 48 h fasted mice, approximately 10% of the control animals were unable to recover and died (Figure 9B). However, a higher percentage of the LepR^∆SOCS3^ mice (~45%) were unable to recover from the fasting (Figure 9B). Comparing the mice that survived and those that died, we observed reduced glycemia in the mice that died after prolonged fasting (Figure 9C). Thus, the ability to sustain glycemia during fasting was probably decisive in increasing their chances of survival. To maintain glycemia during prolonged fasting, it is imperative that hepatic gluconeogenesis works properly [52,56]. Pyruvate tolerance tests were performed, and LepR^∆SOCS3^ mice exhibited impaired hepatic gluconeogenesis in the fed state and after 24 h or 48 h of fasting (Figure 9D–F). Of note, the longer the fasting time, the worse the gluconeogenesis capacity of LepR^∆SOCS3^ mice (Figure 9G). Thus, the inability to perform gluconeogenesis, when necessary, explains the intolerance to fasting and the high mortality rate of the LepR^∆SOCS3^ mice during prolonged food deprivation.

### 2.5. Possible Alterations in the Sympathetic Nervous System Explain the Impaired Gluconeogenesis and Counterregulatory Response of LepR^∆SOCS3^ Mice

During fasting or hypoglycemia, several counterregulatory hormones are secreted, whereas a decrease in plasma insulin concentrations is observed [49,52,53,55]. To determine the physiological mechanisms behind the impaired gluconeogenesis of LepR^∆SOCS3^ mice during fasting, the concentrations of several counterregulatory hormones were assessed. Fasting led to an increase in corticosterone and growth hormone (GH) concentrations (Figure 10A,B). However, no differences between the control and the LepR^∆SOCS3^ mice were observed either in the fed state or during fasting (Figure 10A,B). In contrast, serum insulin concentrations decreased during fasting without differences between the groups (Figure 10C). Since similar hormonal responses were observed, we investigated the role of the autonomic nervous system in controlling hepatic gluconeogenesis in LepR^∆SOCS3^ mice. To block the parasympathetic nervous system, pyruvate was co-infused with atropine, a muscarinic receptor antagonist. Similar to our previous results, LepR^∆SOCS3^ mice showed a lower gluconeogenic capacity either in the fed state or in fasted animals, despite the blockade of the parasympathetic nervous system (Figure 10D,E). Conversely, the blockade of the sympathetic nervous system by the co-infusion of α and β antagonists made the differences between control and LepR^∆SOCS3^ mice disappear, since a similar area under the curve in the pyruvate tolerance tests was observed at the fed state or during fasting (Figure 10F,G).

To additionally determine the capacity of LepR^∆SOCS3^ mice to prevent glucopenia, the counterregulatory response to the infusion of 2-deoxy-D-glucose (2DG) was evaluated. Confirming the reduced capacity to prevent glucopenia, LepR^∆SOCS3^ mice showed a blunted counterregulatory response to 2DG (Figure 10H). Furthermore, the difference in the counterregulatory response between the control and the LepR^∆SOCS3^ mice was no longer observed when the sympathetic nervous system was pharmacologically blocked (Figure 10I). Gut motility is mainly regulated by the parasympathetic nervous system [57,58]. Gut motility was evaluated in the experimental animals and no differences between control (0.72 ± 0.04 arbitrary units) and LepR^∆SOCS3^ (0.73 ± 0.03 arbitrary units; *p* = 0.8694) mice were observed. Taken together, LepR^∆SOCS3^ mice probably exhibit alterations in the activity of the sympathetic nervous system leading to defects in their capacity to maintain glycemia during conditions of glucopenia. In contrast, no evidence of alterations in the activity of the parasympathetic nervous system was observed in LepR^∆SOCS3^ mice.

## 3. Discussion

Some degree of leptin resistance is frequently observed in obese individuals [5,6]. Considering that the activation of LepR in hypothalamic neurons reduces food intake and may increase energy expenditure and fat oxidation [2,7], leptin sensitizing therapies will likely produce robust beneficial effects in the prevention and treatment of obesity [6,7,26]. However, several neurocircuits activated by leptin lead to increased activity of the sympathetic system [20,59,60,61] and inhibition of hepatic glucose production [16,49,52,53]. Therefore, leptin sensitizing treatments may cause side effects, especially regarding the regulation of these autonomic functions. Accordingly, central leptin action increases the release of endogenous agonists of the melanocortin-4 receptor (MC4R), such as the α-melanocyte-stimulating hormone [7,62]. Although the activation of MC4R produces satiety, which is interesting from the point of view of treating obesity, it also increases sympathetic activity and blood pressure, which limits the clinical use of MC4R agonists [63,64]. In the present study, we investigated whether increased leptin sensitivity produces beneficial or deleterious effects in aging mice. As expected, increased leptin sensitivity improved energy and glucose metabolism in middle-aged animals. However, these beneficial consequences were insufficient to improve health and longevity, since LepR^∆SOCS3^ mice exhibited a high mortality rate, morphofunctional abnormalities in the cardiovascular system and impaired ability to maintain blood glucose concentrations in situations of glucopenia.

LepR^∆SOCS3^ mice were well-characterized in previous studies and showed increased leptin sensitivity, resistance against diet-induced leptin resistance and improved energy and glucose metabolism in different situations [27,28,29,30,31,32]. Thus, the reduced body weight and improved insulin sensitivity observed in middle-aged LepR^∆SOCS3^ mice were expected results. Aging increases SOCS3 expression in the hypothalamus [34] and aged animals exhibit central leptin resistance [33,35,36,37,38,39]. Since middle-aged LepR^∆SOCS3^ mice remained responsive to the anorexigenic effect of leptin, our findings suggest that the SOCS3 protein plays a key role in the development of aging-induced leptin resistance. However, it is important to mention that SOCS3 also modulates the activity of other cytokines besides leptin [65]. Consequently, we cannot rule out the possibility that part of the phenotype exhibited by LepR^∆SOCS3^ mice is not necessarily associated with leptin action but is caused by changes in the sensitivity to other hormones. This confounding factor should be considered when analyzing our findings.

Surprisingly, LepR^∆SOCS3^ mice showed increased mortality, precluding us to study animals older than 16 months of life. The causes of death were undetermined, but our general impression was that LepR^∆SOCS3^ mice were more fragile and appeared to be in worse physical condition than age-matched control animals. Since the improved metabolic parameters of LepR^∆SOCS3^ mice could not explain their increased mortality, we investigated possible alterations in the cardiovascular system, considering the well-established role of leptin in regulating cardiovascular functions via the sympathetic nervous system [19,20,61,63,66]. Using different cohorts of mice and after analyzing several morphofunctional parameters, both young adult and middle-aged LepR^∆SOCS3^ mice exhibit key alterations in the cardiovascular system. The cardiomyocyte hypertrophy associated with increased myocardial fibrosis help to explain the lower cardiovascular capacity of LepR^∆SOCS3^ mice during an incremental treadmill maximal running test, as compared to control mice. Importantly, these alterations were already detected in young adult mice, indicating that they were independent of aging. Additionally, there is evidence showing that leptin’s effects on blood pressure are also independent of changes in body weight [19,67].

Previous studies have shown that leptin can regulate cardiac functions either indirectly via the sympathetic nervous system [19,20,59,61,63,64,68] or by acting directly in the heart [69,70,71,72,73,74]. *Lepr* mRNA expression is detected in the heart [69,70,72]. However, immunohistological analysis shows controversial results, in which some studies demonstrate *Lepr* expression in cardiomyocytes [71], while others detect this receptor only in pathological conditions, such as after ischemia and reperfusion [69]. Since our conditional gene deletion was driven by the LepRb-Cre mice [43], we crossed this mouse model with a Cre-dependent tdTomato reporter mice to visualize LepRb-expressing cells in the heart. Using this approach, we only observed LepRb-expressing cells appear in the extracellular matrix and not in the cardiomyocytes. Thus, these findings suggest that SOCS3 deletion did not seem to directly affect cardiomyocytes, therefore, the cardiovascular dysfunctions observed in LepR^∆SOCS3^ mice were likely mediated by the central effects of leptin, which mainly involves the regulation of the activity of the sympathetic nervous system. Although we did not observe significant changes in the basal HR and MAP between the groups, spectral analysis of the SAP and PI waveforms identified increased LF variability and sympathovagal balance in LepR^∆SOCS3^ mice. These alterations can be explained by a suggested higher sympathetic activity [44,45,46,47] and they are in accordance with the well-established stimulatory effect of leptin, as well as the melanocortin system, on the sympathetic nervous system [19,20,59,61,63,64,68].

Using the Langendorff technique of isolated heart perfusion, we observed that young adult LepR^∆SOCS3^ mice exhibit impairment of post-ischemic cardiac functional recovery. In contrast, middle-aged LepR^∆SOCS3^ mice showed increased LVDP during the reperfusion. Although speculative, it is possible that the improved metabolic status of middle-aged LepR^∆SOCS3^ mice, as compared to control aged mice, has contributed to better cardiac function and recovery after ischemia. Nonetheless, severe arrhythmic events were frequently observed in the hearts of middle-aged LepR^ΔSOCS3^ mice during the reperfusion period. Leptin can increase ischemia-related ventricular arrhythmias via sympathetic nerve activation [75]. Considering that obesity, which is a hyperleptinemia condition, represents an important risk factor for heart failure [76], more studies are necessary to investigate the possible association between increased leptin sensitivity and the occurrence of arhythmic events, especially in aged individuals.

Besides the role of leptin in regulating the cardiovascular system, hepatic glucose production is also under the control of leptin-responsive neurons [16,17,18]. Central leptin infusion improves hepatic insulin resistance and markedly inhibits glucose production [17,18]. These effects likely depend on POMC neurons since LepR expression only in POMC-expressing cells is sufficient to suppress hepatic glucose production and normalize blood glucose levels of otherwise LepR-deficient mice [16]. Agouti-related peptide-expressing neurons are also able to regulate systemic insulin resistance and they mediate the glucose-lowering action of leptin [77,78]. Thus, several LepR-expressing neuronal populations in the hypothalamus are able to regulate systemic glucose homeostasis.

Although the inhibition of hepatic glucose production is beneficial for the prevention of type 2 diabetes mellitus, there are situations, such as prolonged fasting, in which the activation of endogenous glucose production is necessary. We observed that LepR^∆SOCS3^ mice are not tolerant to prolonged (48 h) fasting because these animals develop hypoglycemia leading to a high mortality rate. Using pyruvate tolerance tests, we observed that LepR^∆SOCS3^ mice exhibit suppressed hepatic gluconeogenesis. Furthermore, the ability of LepR^∆SOCS3^ mice to convert pyruvate to glucose decreased over the fasting period, which helps to explain their fasting-induced hypoglycemia. The impaired capacity of LepR^∆SOCS3^ mice to recover from glucoprivic situations was further confirmed by the decreased counterregulatory response exhibited after 2DG injection. The concentrations of some counterregulatory hormones were determined in fed and fasted animals and no significant differences were observed. In addition, the co-infusion of atropine, a parasympathetic blocker, did not prevent the differences between the control and the LepR^∆SOCS3^ mice in blood glucose levels during the pyruvate tolerance tests. Thus, neither neuroendocrine defects nor the parasympathetic nervous system seems to account for the defects in glucose production exhibited by LepR^∆SOCS3^ mice. In contrast, co-infusion of α and β blockers with either pyruvate or 2DG was sufficient to abolish the differences between the groups. Therefore, the suppressed glucose production of LepR^∆SOCS3^ mice likely relies on alterations in the activity of the sympathetic nervous system.

Interestingly, SOCS3 ablation in steroidogenic factor-1 (SF1)-expressing cells, which includes the ventromedial nucleus of the hypothalamus (VMH), reduces blood glucose levels in fed and fasted mice, even though the animals do not develop hypoglycemia [79]. The prevention of glutamatergic neurotransmission from VMH neurons also reduces fasting glycemia and the counterregulatory response to hypoglycemia [80]. Thus, changes in leptin sensitivity in VMH neurons are possibly underlying the reduced capacity of LepR^∆SOCS3^ mice to recover from glucoprivic situations. Furthermore, since the blockade of the sympathetic nervous system prevents the defects exhibited by LepR^∆SOCS3^ mice, our findings are in accordance with the literature indicating that VMH neurons regulate glucose homeostasis predominantly via the sympathetic nervous system [52,68].

In conclusion, we provided evidence that mice carrying SOCS3 ablation in LepR-expressing cells retain leptin sensitivity and present improved energy and glucose homeostasis during aging. However, these animals display a high mortality rate, and they develop cardiomyocyte hypertrophy, increased myocardial fibrosis, reduced cardiovascular capacity and impaired cardiac functional recovery after ischemia. LepR^∆SOCS3^ mice also show fasting-induced hypoglycemia, which is caused by a profound reduction in hepatic glucose production. These neurovegetative dysfunctions are likely mediated by alterations in the sympathetic nervous system, compromising their health and longevity. In the context of therapies that seek to chronically increase leptin sensitivity for the treatment of obesity, these potentially negative aspects need to be considered and carefully evaluated in pre-clinical and clinical studies.

## 4. Materials and Methods

### 4.1. Animals

To induce SOCS3 ablation in cells that express the LepRb, the LepRb-IRES-Cre strain (Strain #008320, The Jackson Laboratory, Bar Harbor, ME, USA) was initially bred with SOCS3^flox/flox^ mice (Strain #010944, The Jackson Laboratory). After successive generations, mice homozygous for the LepRb-Cre allele and heterozygous for the loxP-flanked *Socs3* allele were continuously bred to generate the experimental animals. Mice carrying SOCS3 ablation in LepR-expressing cells were homozygous for both loxP-flanked *Socs3* and LepRb-Cre alleles (hereafter named LepR^∆SOCS3^), whereas the control group was composed of littermate mice carrying only the LepRb-Cre allele. LepRb-reporter mouse was produced as previously described [42]. In the experiments, we used either young adult (approximately 3-month-old) or middle-aged (approximately 15-month-old) mice. Only male mice were used in the experiments. Animals were maintained in standard conditions of light (12 h light/dark cycle; lights on at 8:00) and temperature (22 ± 2 °C). Mice received a regular rodent chow (2.99 kcal/g; 9.4% kcal derived from fat; Nuvilab CR-1, Quimtia, Brazil) and filtered water ad libitum, except when indicated. The animal procedures were previously approved by the Ethics Committee on the Use of Animals of the Institute of Biomedical Sciences at the University of São Paulo (protocol number 12/2013).

### 4.2. Energy and Glucose Homeostasis

The body weight of control and LepR^∆SOCS3^ mice were determined weekly from two months of age to approximately 15 months of life. After that, mice were single-housed for acclimation, followed by determination of food intake for 4 consecutive days. VO_2_, CO_2_ production, ambulatory activity (by infrared sensors) and respiratory quotient (CO_2_ production/O_2_ consumption) were determined using the Oxymax/Comprehensive Lab Animal Monitoring System (Columbus Instruments, Athens, OH, USA) for approximately 7 days. The data from the first 3 days of analysis were discarded because we considered the acclimatization period. The results used for each animal were the average of the analyzed days. Subsequently, glucose tolerance tests and insulin tolerance tests were performed. Initially, food was removed from cage 4 h before each test. After the evaluation of basal glucose level (time 0), mice received an intraperitoneal (i.p.) injection of 1 IU insulin/kg or 1 g glucose/kg in middle-aged mice or 2 g glucose/kg in young adult mice, followed by serial determinations of glycemia. During the entire follow-up period, any death of the animals was recorded in order to determine the survival curve.

### 4.3. Tissue Collection and Processing

Mice were euthanized by decapitation after 4 h of food deprivation. The masses of the subcutaneous, perigonadal and retroperitoneal fat pads were determined. Transverse heart sections were fixed in 4% paraformaldehyde overnight and stored in 70% ethanol. After dehydration, samples were embedded in paraffin and sectioned into 5 μm-thick slices. Transverse heart sections were stained with picrosirius red to determine myocardial collagen deposition (*n* = 4–5 hearts per group). The cardiac fibrosis area was measured, and the results are presented as relative fibrosis area in relation to total cardiac area. Transverse heart sections were stained with hematoxylin and eosin to quantify the relative cardiomyocyte transverse diameter (*n* = 50 cardiomyocytes per heart containing visible cell limits and central nuclei, *n* = 4–5 hearts per group). Images were obtained using a light microscope (Nikon Eclipse E600 microscope, Tokyo, Japan) and Nikon Software Nis-Elements AR (Hlinsko, Czech Republic) and measured using ImageJ software. Gastrointestinal motility was assessed by using the charcoal method as described previously [58].

### 4.4. Incremental Treadmill Maximal Running Test

Mice were adapted on the treadmill for 3 consecutive days at 5 m/min speed for 10 min. This speed was maintained to avoid a training effect. Then, mice performed an incremental treadmill maximal running test. Starting at 10 m/min, the speed was increased by 3 m/min every 3 min. The test stopped when the animal was incapable to maintain the running speed for more than 5 s. The time required to reach fatigue and the mean velocity of the test were determined.

### 4.5. Leptin Sensitivity

The feeding response to leptin was determined in mice that received an i.p. injection of phosphate-buffered saline (PBS) or mouse recombinant leptin (2.5 µg/g body weight; from Dr. A.F. Parlow, NHPP, Torrance, CA, USA) 2 h before the dark phase, and their food intake was recorded 24 h later. All animals received both injections, so each mouse acted as its own control, and we compared the food intake after PBS and leptin administration.

### 4.6. Constant Flow Langendorff Preparation

Young adult and middle-aged mice were euthanized by decapitation 10 min after an i.p. heparin sodium injection (150 IU). Immediately after emergency thoracotomy and rapid cardiac arrest by superfusion with ice-cold isotonic saline, the hearts were rapidly attached to the Langendorff apparatus (ADInstruments, Castle Hill, NSW, Australia) via aortic cannulation (21-gauge needles) and then were retrogradely perfused under a constant perfusion flow (3 mL/min). The composition of Krebs-Henseleit (KH) perfusate was as follows (mM): NaCl (118), KCl (4.7), CaCl_2_ (1.75), MgSO_4_ (1.66), NaHCO_3_ (24.88), KH_2_PO_4_ (1.18), dextrose 2 g/L, bi-distilled water and equilibrated with 95% O_2_ and 5% CO_2_, warmed at 37 ± 1 °C and pH of 7.4.

The KH buffer was prepared at the time of the experiment and filtered (47 mm Swinnex^®^, Merck KGaA, Darmstadt, Germany; membrane pore 0.22 mM) immediately before infusion [48]. The hearts from all groups were made to undergo a stabilization period of 30 min and then were subjected to global ischemia by stopping the perfusion pump. The flow was returned after 20 min, and the hearts were reperfused for 60 min. A short cannula was passed via the mitral valve and pierced through the apex of the left ventricle (LV) to vent Thebesian drainage. Shortly thereafter, a water-filled polyvinyl chloride film balloon was inserted into the left ventricle through an incision in the left atrial appendage, via the mitral valve, and secured by a ligature [48]. The ventricular balloon was connected via fluid-filled tubing to a pressure transducer coupled to software (LabChart 8.0) for assessment of ventricular performance during full experiment. The balloon was filled to yield a left ventricular end-diastolic pressure of 5–10 mmHg during the initial 10 min of stabilization; after which, the pressure was not further adjusted. To assess left ventricular systolic function, LVDP, +dP/dt, −dP/dt and HR were measured. All these parameters were monitored continuously by a recording system (PowerLab Chart 8-Lab, ADInstruments, Castle Hill, NSW, Australia).

### 4.7. Hemodynamic Recordings and Spectral Analysis

The animals were anesthetized with a mixture of ketamine (100 mg/kg) and xylazine (7 mg/kg; i.p.) and were implanted with a Micro-Renathane tubing catheter into the femoral artery for direct AP measurement. The muscle and skin layers were separately sutured, and the animals were treated with antibiotic protection (160,000 U kg^−1^ benzylpenicillin, intramuscular). On the next day, MAP and HR were acquired in non-anesthetized unrestrained mice in a time frame of 3 h. Baseline MAP, SAP and PI signals were recorded by connecting the arterial catheter to a pressure transducer (MLT0699, ADInstruments), which was coupled to a preamplifier (FE221 Bridge Amp, ADInstruments), and to a computer data acquisition system (model PowerLab 8SP, ADInstruments).

Spectral analysis of the SAP and PI waveforms was performed using the software CardioSeries (https://www.danielpenteado.com/; CardioSeries V2.4). A stable 10 min of SAP and PI records without artifacts or large sudden blood pressure changes of each animal were processed with computer software (LabChart Pro, ADInstruments) that employs an algorithm to detect beat-to-beat inflection points in the pulsatile arterial pressure signal. Beat-by-beat series with SAP and PI values were generated and loaded into the CardioSeries V2.4 software in order to perform cardiovascular variability within time and frequency domain analysis [81].

SAP and PI power spectral density were estimated by a fast Fourier transform algorithm for time series. Using 10 Hz of interpolation rate, beat-by-beat series were divided into half-overlapping sequential sets with 512 points. The spectra of SAP and PI were integrated into LF (0.2–0.75 Hz) and HF (0.75–3 Hz) bands. The results were expressed in absolute (ms^2^) and normalized units, obtained by calculating the percentage of LF and HF power with regard to the total power of the spectrum minus the very low-frequency band (<0.2 Hz) [82]. Sympatho-vagal balance, the LF/HF ratio of PI variability, was also calculated [44,81].

### 4.8. Glucose Responses to Fasting, Pyruvate and 2-Deoxy-D-Glucose

Blood glucose levels were determined after 24 h and 48 h of fasting using a glucometer (One Touch Ultra; Johnson & Johnson, New Brunswick, NJ, USA). Glycemia at the fed state was also determined in mice without access to food for 4 h to standardize the postprandial state. Enzyme-linked immunosorbent assays were used to determine the serum concentrations of corticosterone (Arbor Assays, Ann Arbor, MI, USA), GH (Millipore, Billerica, MA, USA) and insulin (Crystal Chem, Downers Grove, IL, USA). These assays have a limit of detection determined as 20.9 pg/mL, 0.07 ng/mL and 0.1 ng/mL, respectively, and intra- and inter-assay coefficients of variability ≤ 10%. Pyruvate tolerance tests (0.5 g pyruvate/kg; Sigma-Aldrich, St. Louis, MO, USA) were performed in mice at the fed state (4 h of food removal) or after 24 h and 48 h of fasting. The changes in blood glucose levels induced by an i.p. injection of 2DG (0.5 g/kg; Sigma-Aldrich) were also determined. The participation of the autonomic nervous system in the responses to these tests was determined by the co-infused of pyruvate or 2DG with either a muscarinic receptor blocker (3 mg/kg atropine; Sigma-Aldrich) or a combination of α and β receptor blockers (3 mg/kg phentolamine and 0.5 mg/kg propranolol, respectively; Sigma-Aldrich).

### 4.9. Statistical Analysis

The unpaired two-tailed Student’s *t*-test was used to compare data between two groups. Repeated measures two-way ANOVA, followed by Fisher’s least significant difference post-hoc test were used to compare changes along time or in the same animal after different treatments. Log-rank test was used to compare the survival between control and LepR^∆SOCS3^ mice. The GraphPad Prism software (GraphPad, San Diego, CA, USA) was used for the statistical analyses and generation of the graphs. Data were expressed as mean ± standard error of the mean (SEM).

## Figures and Tables

**Figure 1 ijms-23-06484-f001:**
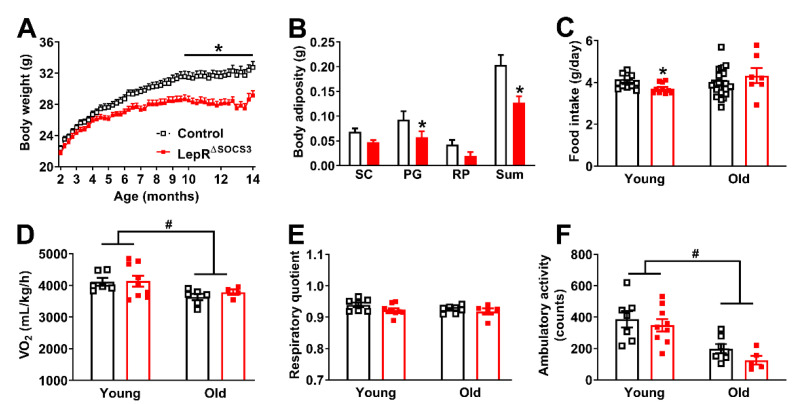
Reduced aging-induced weight gain in LepR^∆SOCS3^ mice. (**A**) Changes in body weight along time in control (*n* = 26–27) and LepR^∆SOCS3^ (*n* = 12–24) mice. (**B**) Masses of the subcutaneous (SC), perigonadal (PG) and retroperitoneal (RP) fat pads and the sum of these deposits in control (black column; *n* = 11) and LepR^∆SOCS3^ (red column; *n* = 10) mice. (**C**) Food intake in young adult (3-month-old; *n* = 10–11/group) and middle-aged (15-month-old; *n* = 7–26/group) mice. (**D**–**F**) Oxygen consumption (VO_2_), respiratory quotient and ambulatory activity in young adult (*n* = 7–9/group) and middle-aged (*n* = 5–7/group) mice. Mean ± SEM. * *p* < 0.05 control vs. LepR^∆SOCS3^ (post-hoc test two-way ANOVA). # *p* < 0.05 age effect (two-way ANOVA).

**Figure 2 ijms-23-06484-f002:**
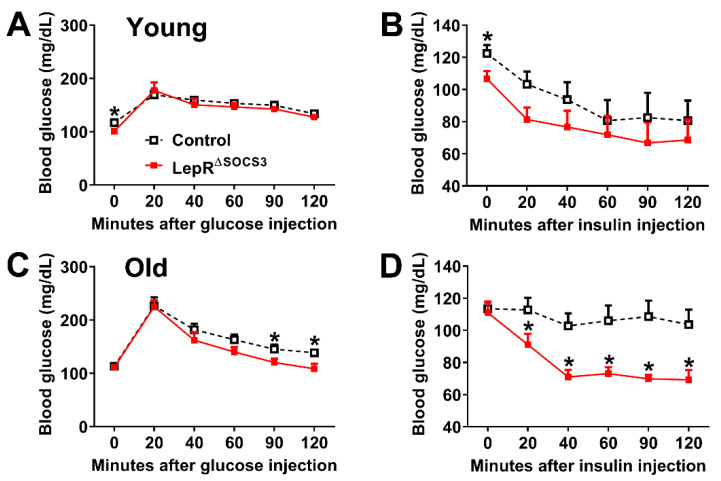
Improved glucose homeostasis in middle-aged LepR^∆SOCS3^ mice. (**A**,**B**) Changes in blood glucose concentrations during a glucose tolerance test and an insulin tolerance test in young adult (3-month-old) control (*n* = 11) and LepR^∆SOCS3^ (*n* = 10) mice. (**C**,**D**) Changes in blood glucose concentrations during a glucose tolerance test and an insulin tolerance test in middle-aged (15-month-old) control (*n* = 21–26) and LepR^∆SOCS3^ (*n* = 8–9) mice. Mean ± SEM. * *p* < 0.05 control vs. LepR^∆SOCS3^ (post-hoc test repeated measures two-way ANOVA).

**Figure 3 ijms-23-06484-f003:**
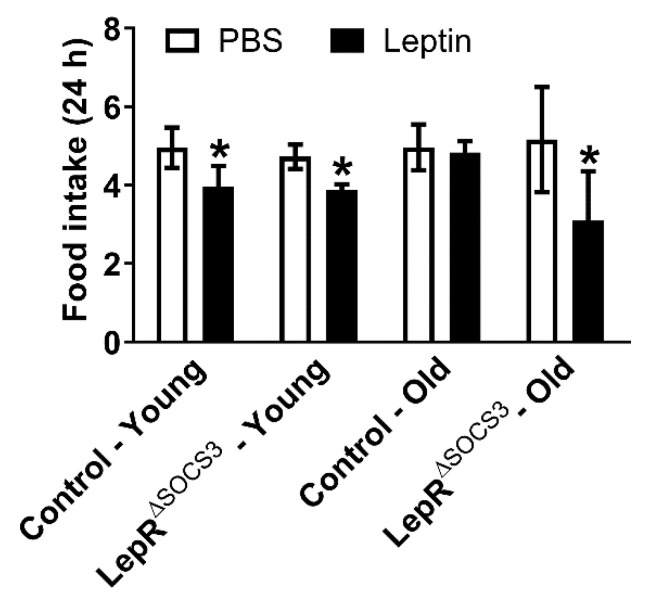
LepR^∆SOCS3^ mice are protected against aging-induced leptin resistance. Twenty-four hour food intake after an i.p. injection of PBS or leptin (2.5 µg/g body weight) in young adult (3-month-old) control (*n* = 6) and LepR^∆SOCS3^ (*n* = 7) mice and in middle-aged (15-month-old) control (*n* = 7) and LepR^∆SOCS3^ (*n* = 4) mice. Mean ± SEM. * *p* < 0.05 control vs. LepR^∆SOCS3^ (post-hoc test repeated measures two-way ANOVA).

**Figure 4 ijms-23-06484-f004:**
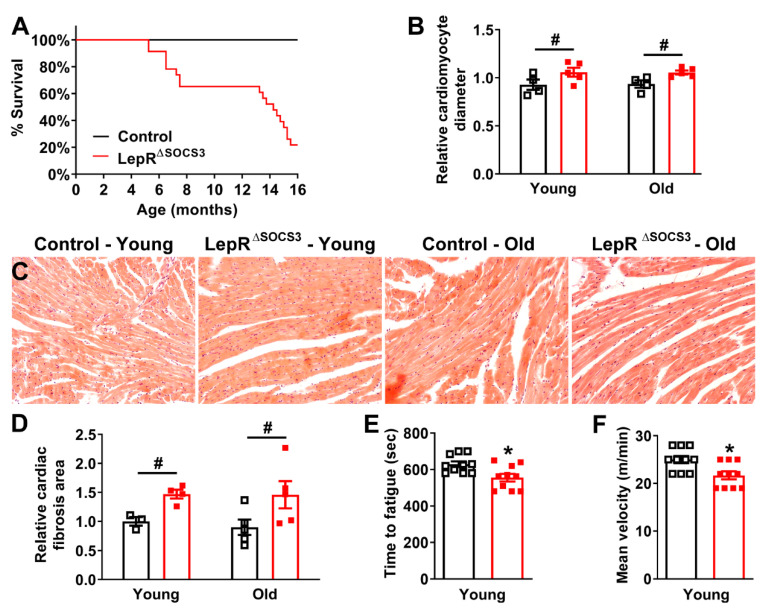
Cardiac abnormalities and decreased survival rate in aging LepR^∆SOCS3^ mice. (**A**) Survival curve of control (black square; *n* = 27) and LepR^∆SOCS3^ (red square; *n* = 23) mice. (**B**) Relative cardiomyocyte transverse diameter in young adult (3-month-old; *n* = 4–5/group) and middle-aged (15-month-old; *n* = 4–5/group) mice. (**C**) Representative transverse heart sections stained with hematoxylin and eosin of young adult or middle-aged control and LepR^∆SOCS3^ mice. (**D**) Relative cardiac collagen area in young adult (*n* = 3–4/group) and middle-aged (*n* = 5/group) mice. (**E**,**F**) Time required until fatigue and mean velocity during an incremental treadmill maximal running test in young adult control (*n* = 10) and LepR^∆SOCS3^ (*n* = 10) mice. Mean ± SEM. * *p* < 0.05 control vs. LepR^∆SOCS3^ (*t*-test). # *p* < 0.05 SOCS3 ablation effect (two-way ANOVA).

**Figure 5 ijms-23-06484-f005:**
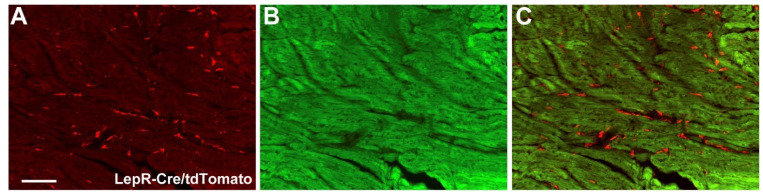
LepRb is not expressed in cardiomyocytes. (**A**–**C**) Epifluorescence photomicrographs showing the expression of tdTomato protein (red straining) in the heart of a LepR-reporter mouse. Note that tdTomato is expressed in the tissue matrix, but not in cardiomyocytes. Scale bar = 100 µm.

**Figure 6 ijms-23-06484-f006:**
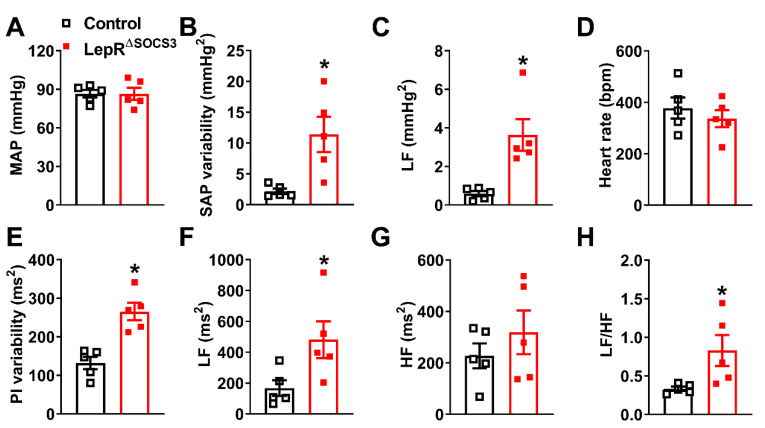
Increased sympathetic activity to the vasculature and heart in LepR^∆SOCS3^ mice. (**A**–**C**) Mean arterial pressure (MAP), systolic arterial pressure (SAP) variability and low-frequency (LF) variability of the SAP in approximately 8-month-old control (*n* = 5) and LepR^∆SOCS3^ (*n* = 5) mice. (**D**–**G**) Heart rate (HR), pulse interval (PI) variability and LF and high-frequency (HF) variability of the PI. (**H**) Sympathovagal balance (LF/HF ratio) of PI variability. Mean ± SEM. * *p* < 0.05 control vs. LepR^∆SOCS3^ (*t*-test).

**Figure 7 ijms-23-06484-f007:**
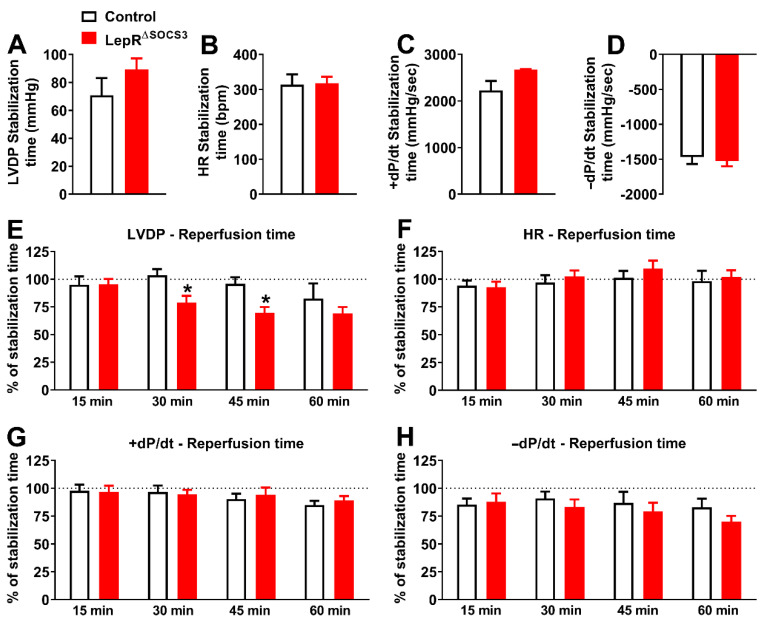
Impairment of post-ischemic cardiac functional recovery in young adult LepR^ΔSOCS3^ mice. (**A**–**D**) Baseline cardiac function including left ventricular developed pressure (LVDP), heart rate (HR), first derivative of the positive (+dP/dt) and negative (−dP/dt) ventricular pressure in young adult control (*n* = 6) and LepR^∆SOCS3^ (*n* = 5) mice. (**E**–**H**) Temporal analysis of cardiac parameters at 15, 30, 45 and 60 min of reperfusion in young adult mice. Mean ± SEM. * *p* < 0.05 control vs. LepR^∆SOCS3^ (post-hoc test repeated measures two-way ANOVA).

**Figure 8 ijms-23-06484-f008:**
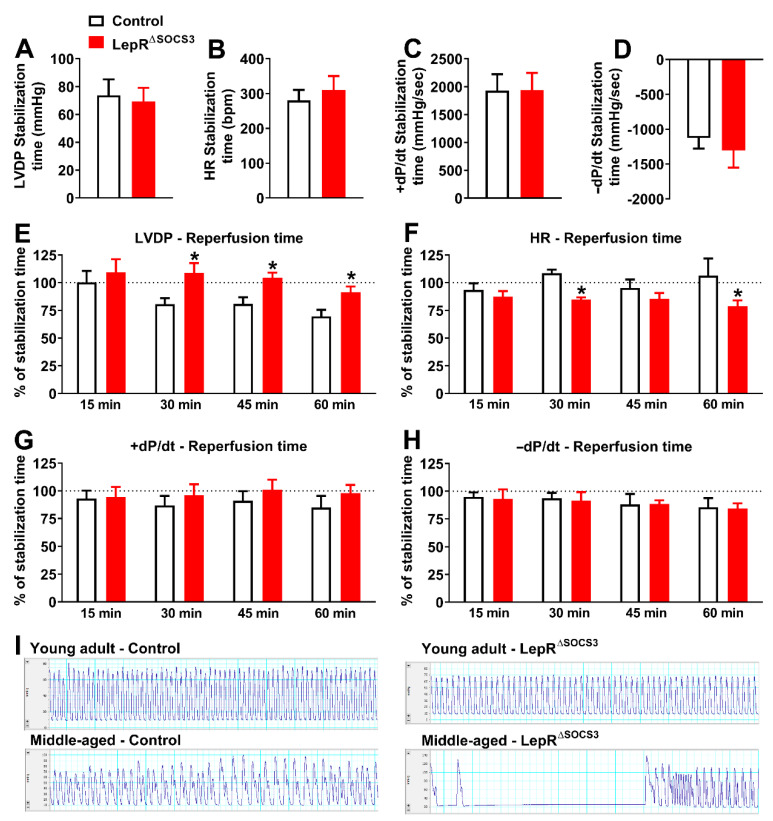
Substantial arhythmic events during the reperfusion period in middle-aged LepR^ΔSOCS3^ mice despite an apparent improvement in the post-ischemic cardiac functional recovery. (**A**–**D**) Baseline cardiac functions including left ventricular developed pressure (LVDP), heart rate (HR), first derivative of the positive (+dP/dt) and negative (−dP/dt) ventricular pressure in middle-aged control (*n* = 7) and LepR^∆SOCS3^ (*n* = 6) mice. (**E**–**H**) Temporal analysis of cardiac parameters at 15, 30, 45 and 60 min of reperfusion in middle-aged mice. (**I**) Representative recordings during the reperfusion period showing a robust arhythmic event in middle-aged LepR^ΔSOCS3^ mice. Mean ± SEM. * *p* < 0.05 control vs. LepR^∆SOCS3^ (post-hoc test repeated measures two-way ANOVA).

**Figure 9 ijms-23-06484-f009:**
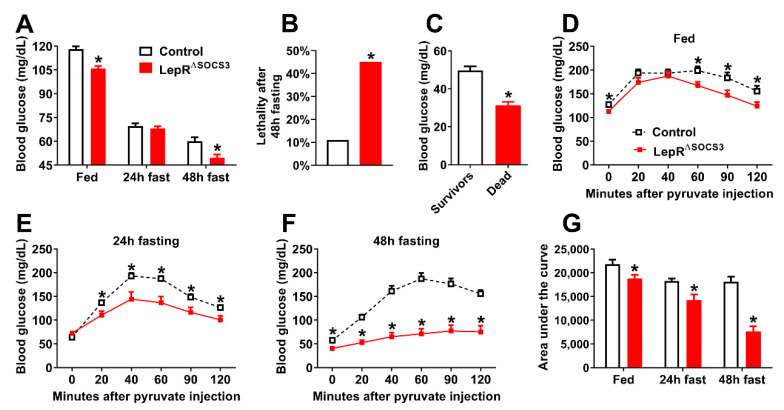
LepR^∆SOCS3^ mice exhibit fasting-induced hypoglycemia that is associated with reduced gluconeogenesis. (**A**) Blood glucose concentrations at the fed state or after 24 h or 48 h fasting in young adult control (*n* = 40) and LepR^∆SOCS3^ (*n* = 32) mice. (**B**) Percentage of mice that died after a 48 h fasting period in control and LepR^∆SOCS3^ mice. (**C**) Blood glucose concentrations in mice that died (*n* = 20) or survived (*n* = 27) the 48 h fasting period (including the refeeding period). (**D**–**F**) Blood glucose concentrations during pyruvate tolerance tests in young adult control (*n* = 20–21) and LepR^∆SOCS3^ (*n* = 8–11) mice at the fed state or during 24 h or 48 h fasting. (**G**) Area under the curve of the pyruvate tolerance tests in control and LepR^∆SOCS3^ mice at the fed state or during 24 h or 48 h fasting. Mean ± SEM. * *p* < 0.05 control vs. LepR^∆SOCS3^ (post-hoc test two-way ANOVA or *t*-test).

**Figure 10 ijms-23-06484-f010:**
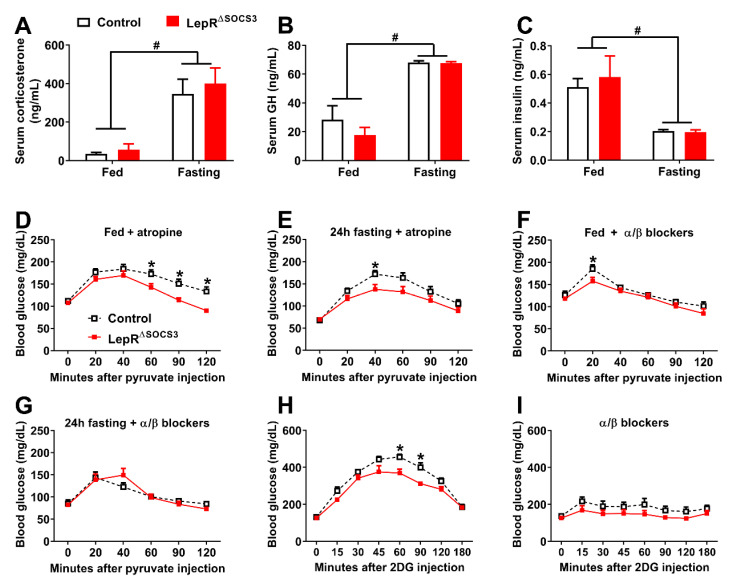
Possible alterations in the sympathetic nervous system explain the impaired gluconeogenesis and counterregulatory response of LepR^∆SOCS3^ mice. (**A**–**C**) Serum corticosterone, GH and insulin concentrations in the fed state or after 24 h fasting in young adult control (*n* = 5–8) and LepR^∆SOCS3^ (*n* = 4–8) mice. (**D**–**G**) Blood glucose levels during pyruvate tolerance tests in young adult control (*n* = 6–14) and LepR^∆SOCS3^ (*n* = 5–13) mice at the fed state or during 24 h fasting in combination with the co-infusion of atropine (**D**,**E**) or α/β blockers (**F**,**G**). (**H**,**I**) Blood glucose levels show the counterregulatory response induced by 2DG injection in young adult control (*n* = 7–9) and LepR^∆SOCS3^ (*n* = 7–9) mice. Mean ± SEM. # *p* < 0.05 fasting effect (two-way ANOVA). * *p* < 0.05 control vs. LepR^∆SOCS3^ (post-hoc test repeated measures two-way ANOVA).

## Data Availability

The data that support the findings of this study are available from the corresponding author upon reasonable request.

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
