# Peer review of "SOCS3 Ablation in Leptin Receptor-Expressing Cells Causes Autonomic and Cardiac Dysfunctions in Middle-Aged Mice despite Improving Energy and Glucose Metabolism"

_ijms, 2022, doi:10.3390/ijms23126484_

Round 1

Reviewer 1 Report

The authors investigated the potential beneficial and deleterious effects of increased leptin sensitivity using leptin receptor cell-specific SOCS3 KO mice. They found that the LepRΔSOCS3 mice displayed reduced body weight gain, aging-induced leptin resistance, and improved glucose homeostasis. However, these mice has very high mortality rate and exhibit cardiac malfunction. In addition, LepRΔSOCS3 mice also showed impaired gluconeogenesis and reduced capacity to prevent glucopenia. The finding is interesting regarding diverse neurovegetative effect of increased leptin sensitivity. I have 3 comments.

Comments:

  1. Results in Figure 3 and supplementary figure 1 are somewhat conflicting. They showed increased Npy and decreased Pomc/Crh gene expressions level in the middle-aged LepR-SOCS3 mice, but food intake was reduced. The interpretation of these results is kind of insufficient.
  2. Figure 8A-C; there is no difference in counterregulatory hormones between the genotypes, so it seems not reasonable to conclude that there is impaired in the counterregulatory response in KO. In this regard, what about changes in the amount of glucagon induced by fasting?
  3. It would be helpful if they describe their breeding strategy more detail. They described that they used littermates but they used homo cre together with homo floxed mice. It would be difficult to get KO and WT (without floxed mice) littermates.

Author Response

  1. Results in Figure 3 and supplementary figure 1 are somewhat conflicting. They showed increased Npy and decreased Pomc/Crh gene expressions level in the middle-aged LepR-SOCS3 mice, but food intake was reduced. The interpretation of these results is kind of insufficient.

RESPONSE: We would like to thank the reviewer for the time spent evaluating our manuscript and their constructive comments and suggestions. In the current version of the manuscript, the results regarding the hypothalamic gene expression (mentioned by the reviewer as supplementary figure 1) were not included because we believe that these results do not contribute with the paper. Anyhow, we found increased Npy and decreased Pomc/Crh gene expressions in the hypothalamus of middle-aged LepRΔSOCS3 mice, compared to control mice, although middle-aged LepRΔSOCS3 mice retained leptin sensitivity (Figure 3). The reviewer is correct in pointing out that the expected response to increased Npy and decreased Pomc/Crh gene expression would be to induce higher food intake. Actually, middle-aged LepRΔSOCS3 mice show no difference in food intake, as compared to middle-aged control mice (Figure 1C). Although we do not have a definitive explanation for the differences in hypothalamic gene expression, we believe that it is reflecting the leanness of the LepRΔSOCS3 mice (Figure 1A-B). For example, mice lacking melanin-concentrating hormone are hypophagic and lean, despite also presenting decreased Pomc gene expressions in the hypothalamus (Shimada et al., Nature 396:670-674, 1998), which should also theoretically lead to less satiety and increased food intake. Since these data are no longer part of the manuscript, no alterations are necessary in the paper.

  1. Figure 8A-C; there is no difference in counterregulatory hormones between the genotypes, so it seems not reasonable to conclude that there is impaired in the counterregulatory response in KO. In this regard, what about changes in the amount of glucagon induced by fasting?

RESPONSE: We tried to measure glucagon levels, but unfortunately the assay did not work out. Since we also analyzed the levels of several hormones, we ended up running out of samples to reanalyze the serum concentration of glucagon. The counterregulatory response to hypoglycemia is induced by the secretion of counterregulatory hormones like glucagon, GH and glucocorticoids, as well as the suppression of insulin secretion. Additionally, the counterregulatory response is also driven by changes in the autonomic system, more specifically by the activation of the sympathetic nervous system, which rapidly stimulates hepatic glucose production by increasing glycogenolysis and gluconeogenesis, and by the inhibition of the parasympathetic nervous system, that antagonizes with the effects of the sympathetic nervous system. It is possible that defects in the counterregulatory response to hypoglycemia occurs in specific pathways and not necessarily in all endocrine and neuronal responses. For example, we previously have shown that defects in the autonomic nervous system can occur during the counterregulatory response, even though the levels of counterregulatory hormones are normal (Furigo et al., FASEB J 33:11909-11924, 2019). This occurs because the genetic alterations induced in our animal models led to alterations in the circuits that recruit the autonomic nervous system, but not the neuroendocrine responses, mediated, for example, by the hypothalamic-pituitary axis (that controls the secretion of glucocorticoids and GH).

  1. It would be helpful if they describe their breeding strategy more detail. They described that they used littermates but they used homo cre together with homo floxed mice. It would be difficult to get KO and WT (without floxed mice) littermates

RESPONSE: In our previous experience (Zampieri et al., Mol Metab 4:237-245, 2015; Garcia-Galiano et al., JCI insight 2:2017) and also reported by other authors (Leinninger et al., Cell Metab 10:89-98, 2009; Leshan et al., J Neurosci 29:3138-3147, 2009), the expression of the LepRb-Cre allele in homozygosity reproduces more precisely the correct distribution of LepR-expressing cells in the central nervous system, probably because the natural low expression of Lepr gene, which drives Cre expression. Thus, to generate the experimental animals, we bred mice homozygous for the LepRb-Cre allele and heterozygous for the loxP-flanked Socs3 allele. Thus, 25% of the offspring was composed of mice carrying both loxP-flanked Socs3 and LepRb-Cre alleles in homozygosity, which we called LepR∆SOCS3 mice (carrying the cell-specific deletion). Other 25% of the offspring carried only the LepRb-Cre allele in homozygosity and were used as control group, therefore, they were littermates. Finally, 50% of the offspring carried the loxP-flanked Socs3 in heterozygosity (and the LepRb-Cre allele in homozygosity) and were used as breeders.

Reviewer 2 Report

Both the publication and the results obtained in the course of study are very interesting. The potential negative effects of increased leptin sensitivity observed in experiments are really unexpected. I would like to ask Authors to explain some small points.

I know that you wrote that before"This mouse model was characterized and validated in several previous studies, which demonstrated increased  leptin sensitivity and protection against leptin resistance in different situations [27-32]" however Iam sure that a few sentence about that model and expression of SOCS-3 in that model will be necessary for readers. I noticed that in previous experiments you used female premature and pregnant rats, did you observe any difference in leptin sensitivity in male rats?

Please, let me know you used the dose of mouse recombinant leptin 2.5 ug/g of body weight?

Why you you determine GH concentration? 

Please, inform readers about the inter- and intraassay precision CV values and assays sensitivity. 

Did you check leptin concentration after i.p. injection?

Please, change "level" for hormone concentration for "concentration" through the whole paper. 

Author Response

I know that you wrote that before "This mouse model was characterized and validated in several previous studies, which demonstrated increased leptin sensitivity and protection against leptin resistance in different situations [27-32]" however I am sure that a few sentence about that model and expression of SOCS-3 in that model will be necessary for readers. I noticed that in previous experiments you used female premature and pregnant rats, did you observe any difference in leptin sensitivity in male rats?

RESPONSE: We thank the reviewer for time spent evaluating our paper and for the constructive comments that improved the manuscript. We previously demonstrated that SOCS3 deletion in LepR-expressing cells increases leptin sensitivity in pubertal and pregnant female mice (Zampieri et al., Mol Metab 4:237-245, 2015; Bohlen et al., Mol Cell Endocrinol 423:11-21, 2016; Zampieri et al., Physiol Behav 157:109-115, 2016), as well as in male mice consuming either chow or high-fat diet (model of diet-induced obesity) (Pedroso et al., Mol Metab 3:608-618, 2014; Pedroso et al., Endocrinology 157:3901-3914, 2016). As suggested, we added this additional information about the mouse model and the consequences of SOCS3 ablation regarding improving leptin sensitivity in different situations (Page 2):

“For example, former studies have shown that SOCS3 deletion in LepR-expressing cells increases leptin sensitivity in pubertal and pregnant female mice [28,29,31], as well as in male mice consuming either chow or high-fat diet [27,30]. Additionally, the ability of leptin, high-fat diet or pregnancy to upregulate Socs3 mRNA levels in the hypothalamus is completely prevented in LepR∆SOCS3 mice [27,28].”

Please, let me know you used the dose of mouse recombinant leptin 2.5 ug/g of body weight?

RESPONSE: Yes, that is correct. The information about the dose was initially provided in Materials and Methods and now it was also added in the Results and figure legend.

Why you you determine GH concentration?

RESPONSE: Although less acknowledged, GH can also be considered a counterregulatory hormone that is robustly secreted in hypoglycemia (Roth et al., Science 140:987-988, 1963). GH stimulates hepatic glucose production and gluconeogenesis, so it helps to recover from situations of glucopenia. On the other hand, glucocorticoids are classical counterregulatory hormones. Therefore, we determined both GH and corticosterone to have at least two hormones that are secreted during a counterregulatory response under the control of the hypothalamic-pituitary axis. As we mentioned to reviewer #1, we tried to measure glucagon levels, but unfortunately the assay did not work out. Since we also analyzed the levels of several hormones, we ended up running out of samples to reanalyze the serum concentration of glucagon.

Please, inform readers about the inter- and intraassay precision CV values and assays sensitivity.

RESPONSE: The assays used to determine serum concentrations of corticosterone, GH and insulin have a limit of detection determined as 20.9 pg/mL, 0.07 ng/mL and 0.1 ng/mL, respectively, and an intra- and inter-assay coefficients of variability ≤10%. This information was added to the revised manuscript (page 16).

Did you check leptin concentration after i.p. injection?

RESPONSE: We did not. However, we used the same dose of leptin (2.5 ug/g of body weight) in numerous previous publications to determine the acute feeding response to this hormone or the induction of pSTAT3 in the hypothalamus (da Silva et al., Endocrinology 155:4226-4236, 2014; Zampieri et al., Mol Metab 4:237-245, 2015; Bohlen et al., Mol Cell Endocrinol 423:11-21, 2016; Buonfiglio et al., Sci Rep 6:22421, 2016; Pedroso et al., Endocrinology 157:3901-3914, 2016; Ramos-Lobo et al., Neuroscience 365:114-124, 2017; Ramos-Lobo et al., eLife 8:e40970, 2019; Quaresma et al., Endocrinology 162:bqab168, 2021). Thus, we believe that it is well-established that this dose is appropriate to test leptin responsiveness of mice in different situations.

Please, change "level" for hormone concentration for "concentration" through the whole paper

RESPONSE: As suggested by the reviewer, the replaced the word “level” to "concentration".